# The Use of Digital Color Imaging and Machine Learning for the Evaluation of the Effects of Shade Drying and Open-Air Sun Drying on Mint Leaf Quality

**Ewa Ropelewska** [1],*, **Kadir Sabanci** [2] **and Muhammet Fatih Aslan** [2]

1 Fruit and Vegetable Storage and Processing Department, The National Institute of Horticultural Research, 96-100 Skierniewice, Poland
2 Department of Electrical and Electronics Engineering, Karamanoglu Mehmetbey University, Karaman 70100, Turkey
* Correspondence: ewa.ropelewska@inhort.pl

**Abstract:** The objective of this study was to reveal the usefulness of image processing and machine learning for the non-destructive evaluation of the changes in mint leaves caused by two natural drying techniques. The effects of shade drying and open-air sun drying on the ventral side (upper surface) and dorsal side (lower surface) of leaves were compared. Texture parameters were extracted from the digital color images converted to color channels $R$, $G$, $B$, $L$, $a$, $b$, $X$, $Y$, and $Z$. Models based on image features selected for individual color channels were built for distinguishing mint leaves in terms of drying techniques and leaf side using machine learning algorithms from groups of Lazy, Rules, and Trees. In the case of classification of the images of the ventral side of fresh and shade-dried mint leaves, an average accuracy of 100% and values of Precision, Recall, F-Measure, and MCC of 1.000 were obtained for color channels $B$ (KStar and J48 machine learning algorithms), $a$ (KStar and J48), $b$ (KStar), and $Y$ (KStar). The effect of open-air sun drying was greater. Images of the ventral side of fresh and open-air sun-dried mint leaves were completely correctly distinguished (100% correctness) for more color channels and algorithms, such as color channels $R$ and $G$ (J48), $B$, $a$ and $b$ (KStar, JRip, and J48), and $X$ and $Y$ (KStar). The classification of the images of the dorsal side of fresh and shade-dried mint leaves provided 100% accuracy in the case of color channel $B$ (KStar) and $a$ (KStar, JRip, and J48). The fresh and open-air sun-dried mint leaves imaged on the dorsal side were correctly classified at an accuracy of 100% for selected textures from color channels $a$ (KStar, JRip, J48), $b$ (J48), and $Z$ (J48). The developed approach may be used in practice to monitor the changes in the structure of mint leaves caused by drying in a non-destructive, objective, cost-effective, and fast manner without the need to damage the leaves.

**Keywords:** fresh mint leaves; dried leaves; image textures; classification; machine learning

## 1. Introduction

Plants commonly known as mint belong to the genus *Mentha* (Lamiaceae) and are of great economic importance [1]. Mint leaves can be considered an aromatic and medicinal plant material. Mint can alleviate flu, cold, fever, food poisoning, poor digestion, flatulence, motion sickness, rheumatism, hiccups, sinus and throat ailments, and earaches [2]. Consuming mint is highly desirable for its health benefits. However, this herb can only be grown for a certain period of the year. To store mint for a long time, it should be dehydrated [3]. Drying reduces the water content of the plants, inhibits microorganism development and biochemical reactions, and thus allows for preventing degradation, preserving and extending the shelf life of materials. Despite its disadvantages related to, among others, the evaporation of volatile compounds, open-air sun drying is common, mainly in places with plenty of solar radiation [4].

During open-air sun drying, plants are in an open environment directly exposed to solar radiation. This open exposure also results in direct exposure to contamination with insects, dust, or sand particles [5]. Furthermore, it causes discoloration and loss of aroma, which is important for consumers because it can reduce dehydrated-mint quality and quantity. Thermal processing results in numerous reactions affecting the hue. The Maillard reaction, degradation of chlorophyll pigment, and oxidation of ascorbic acid take place. Open-air sun drying causes browning. Furthermore, it destroys the vitamin C content, which is heat- and light-sensitive [3].

Another herb drying method that employs solar energy as a heating source is shade drying. This drying approach protects photosensitive compounds while reducing light-induced chemical reactions such as oxidation [6]. However, the drying time of shade drying is longer than that of sun drying [7]. Additionally, shade drying is a better method for drying herbs since it maintains the integrity of the trichomes [8]. Shade drying also causes less damage to the epidermal surface than sun drying [9]. Nonetheless, due to its low investment cost and high-quality dried products, shade drying, like sun drying, is still popular in rural areas and small enterprises [10].

The leaf classification can be performed based on color, shape, and texture parameters using computer vision [11]. Image processing can be used to detect leaves and extract leaf parameters useful for leaf classification using machine learning algorithms [12]. Machine learning techniques for developing Artificial Intelligence allow computers or machines to learn from experience without explicit programming. The objective of machine learning algorithms is to improve the performance of models using data [13]. Thanks to computer vision-based methods for classifying leaves, various and powerful features can be extracted from leaf images. The leaf features are easy to extract and analyze and can be used for the identification of plants. Manual identification can be vulnerable to human error. Therefore, automated system identification systems are desirable. In the case of mint leaf classification, features from the digital images of the front and back sides of leaves can be taken into account [14]. Intelligent computer models can be very effective in identifying herbs and their properties [15].

In the available literature, there is a lack of information on the presence of classification models based on textures of mint leaf images in the individual color channels *R*, *G*, *B*, *L*, *a*, *b*, *X*, *Y*, and *Z*, computed using image analysis techniques to monitor the effect of natural drying techniques on the changes in the structure of mint leaves. The novelty of the study is to reveal that the changes that occur after leaf drying can be distinguished with computer vision and machine learning applications. This study was aimed at comparing the effect of shade drying and open-air sun drying on the ventral side (upper surface) and dorsal side (lower surface) of mint leaves using image processing and machine learning. The reason for undertaking the research was a need for a more complete understanding of the changes occurring in the image textures of mint leaves caused by different drying techniques to acquire new knowledge and develop the procedure for the detection of these changes.

The contribution of the article is far superior to the current state of the art. The innovative nature of this study is related to the acquisition of new information, not found in the literature, on 1629 texture parameters of fresh and dried mint leaves and the selection of attributes with the highest discriminative power. The article presents a new approach to assessing changes in mint leaves caused by natural drying techniques. The leaf classification performed using models based on selected texture parameters extracted from the digital color images converted to color channels *R*, *G*, *B*, *L*, *a*, *b*, *X*, *Y*, and *Z* built using machine learning algorithms from groups of Lazy, Rules, and Trees is a unique approach to distinguish dried and fresh mint leaves. For the first time, the assessment of the influence of natural drying techniques on the structure of mint leaves was carried out using attributes selected from sets of 1629 image textures from different color channels. Furthermore, monitoring the degree of changes in the structure of leaves under the influence of drying without the need to damage or destroy the leaves can be a problem. The proposed approach may be a non-destructive, objective, cost-effective, and fast practical solution to this problem.

Justification for tackling the problem results from insufficient data in the literature on an experimental and theoretical description of the changes in texture parameters of mint leaf images as a result of the drying. The problem addressed in this study is to reveal, with technological solutions, the textural changes caused by the drying techniques used to preserve the mint leaf for a long time. The proposed solution is to combine the power of computer vision and machine learning, as used in many different agricultural applications today [16,17]. In this way, discrimination systems with non-destructive, non-biased, high-accuracy, and autonomous capabilities can be developed. The proposed solution also does not depend on a single machine learning algorithm or a single color space of images. The robustness of the proposed approach was proven with different machine learning methods and image textures from different color spaces determined experimentally.

## 2. Materials and Methods

### 2.1. Materials

The mint plants were grown in a garden located in central Poland. The experiments were carried out in August. The plants were fully developed. The mint leaves were collected with short petioles. Two or three leaves were taken from a single branch. In total, one hundred and fifty leaves were sampled. The collection of leaves was divided into two parts. The first part of the seventy-five leaves was intended for shade drying. The remaining seventy-five leaves were subjected to open-air sun drying. Before drying, all leaves were imaged on the ventral side (upper surface) and dorsal side (lower surface).

### 2.2. Drying of Mint Leaves

Two drying techniques were applied. The mint leaves were subjected to shade drying and open-air sun drying. Both techniques were natural and used no dryers/dehydrators. The leaves were spread evenly over white sheets of paper placed in boxes with low side walls of about 5 cm. The leaves did not touch each other. Both shade and open-air sun drying experiments started on the same day at 11 a.m.

#### 2.2.1. Shade Drying

The box with mint leaves was placed in a shaded place at room temperature. The leaves were turned over to the upper and lower sides every few hours. The color and structure of the leaves were controlled during the experiment. The leaves were dried for a week until they changed color and dried completely, as assessed organoleptically.

#### 2.2.2. Open-Air Sun Drying

The experiment was carried out on a rainless, windless sunny day. The mint leaves were exposed to direct sunlight. During drying, the leaves were turned on their upper and lower sides every 15 min. When it was noticed that the leaves were beginning to curl, a metal mesh was placed over them for a while to keep the leaves flat. Drying took three hours to completely change the color of the entire leaf surface and to obtain dry leaves.

### 2.3. Image Analysis

One hundred and fifty fresh leaves directly after harvest were subjected to imaging on the ventral and dorsal sides. The same leaves were imaged as the seventy-five shade-dried leaves and seventy-five open-air sun-dried leaves after the drying experiments were completed. Dried mint leaves were also imaged on both sides (ventral and dorsal). Digital color imaging was performed in a dark room using a digital camera and LED (light-emitting diode) as a light source. Leaves were imaged on a white background. Each image contained five leaves. In total, the images of the following samples (classes) were acquired:

-    ventral side of seventy-five fresh mint leaves subjected to shade drying:
-    ventral side of seventy-five shade-dried mint leaves;
-    ventral side of seventy-five fresh mint leaves subjected to open-air sun drying;
-    ventral side of seventy-five open-air sun-dried mint leaves;

- dorsal side of seventy-five fresh mint leaves subjected to shade drying;
- dorsal side of seventy-five shade-dried mint leaves;
- dorsal side of seventy-five fresh mint leaves subjected to open-air sun drying;
- dorsal side of seventy-five open-air sun-dried mint leaves.

The sample images of the ventral sides of fresh and dried mint leaves are shown in Figure 1. The ventral side of fresh leaves was light green. Shade-dried leaves turned dark green. In the case of open-air sun-dried leaves, there was a clear change of color to brown as a result of drying. The exemplary leaves imaged on their dorsal side are presented in Figure 2. On the dorsal side of the leaves, all samples were lighter than on the ventral side. However, the effect of drying was similar. Shade-dried leaves changed color to darker green, whereas under the influence of open-air sun drying the leaves turned brown.

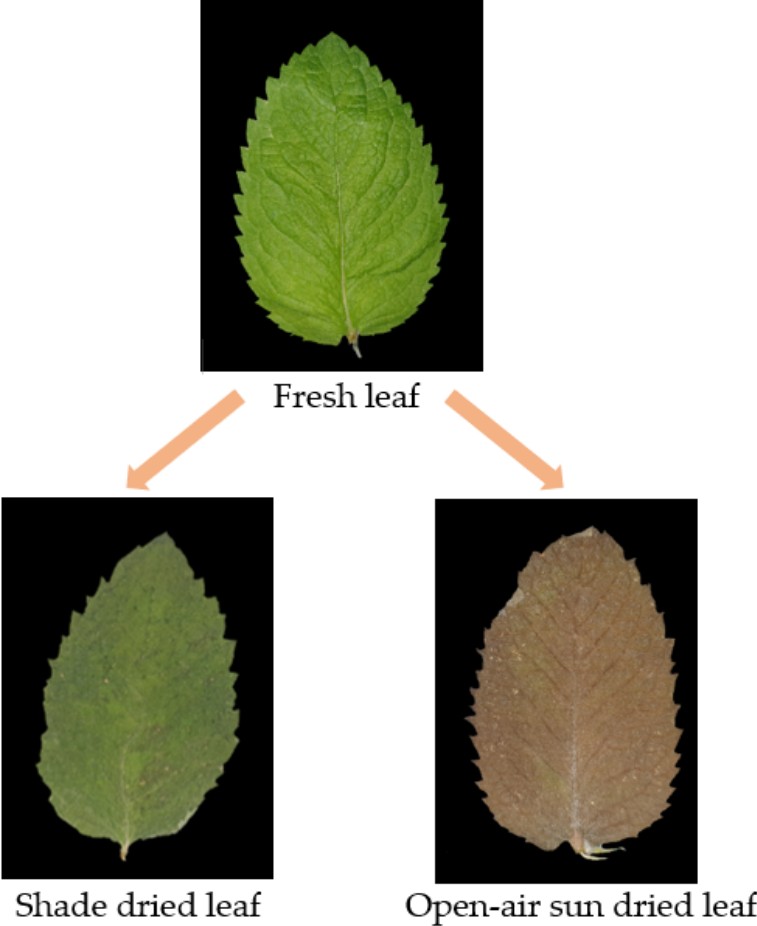

**Figure 1.** The images of the ventral side (upper surface) of fresh and dried mint leaves.

The obtained images were uploaded to a computer with programs for image processing and classification using a USB cable. After changing the image background to black and converting the images to the BMP file format, the image processing was carried out using the MaZda software (Łódź University of Technology, Institute of Electronics, Łódź, Poland) [18–20]. First, the mint leaf images were converted to individual color channels *R*, *G*, *B*, *L*, *a*, *b*, *X*, *Y*, and *Z*. The exemplary images of the ventral side (upper surface) of fresh and dried mint leaves (Figure 3) and the dorsal side (lower surface) of fresh and dried mint leaves (Figure 4) reveal the differentiation of images depending on the color channel and visible changes in leaves caused by drying. The image segmentation was performed to separate the leaves from the black background based on the pixel brightness intensity, and ROIs (regions of interest) were determined. Each ROI contained one whole mint leaf. For each ROI in each of the nine color channels, 181 texture parameters based on the histogram,

gradient map, autoregressive model, run-length matrix, and co-occurrence matrix were extracted. In total, 1629 textures were determined for each leaf image.

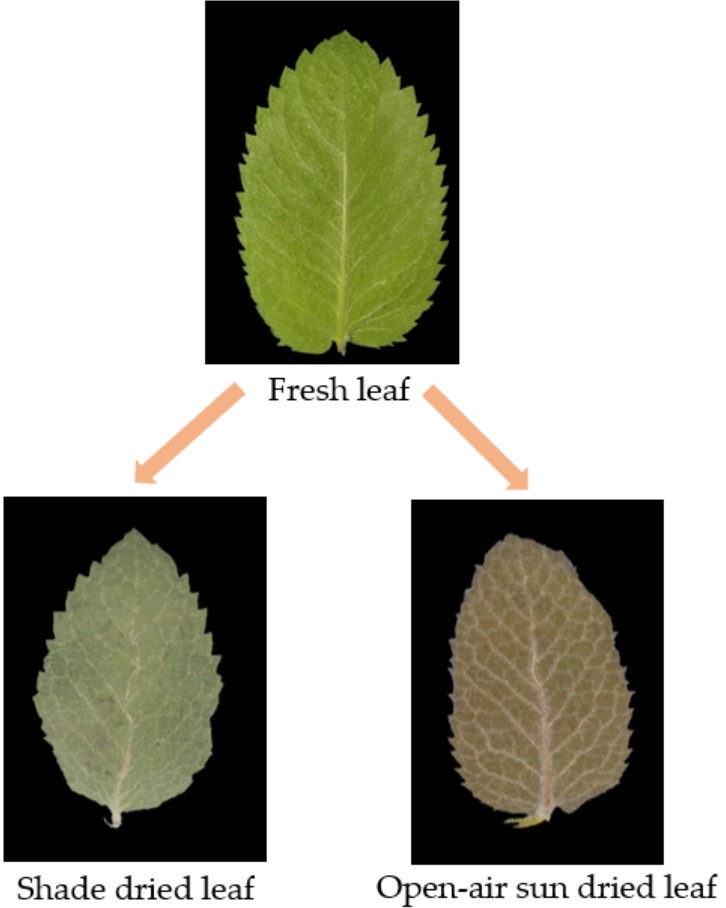

**Figure 2.** The images of the dorsal side (lower surface) of fresh and dried mint leaves.

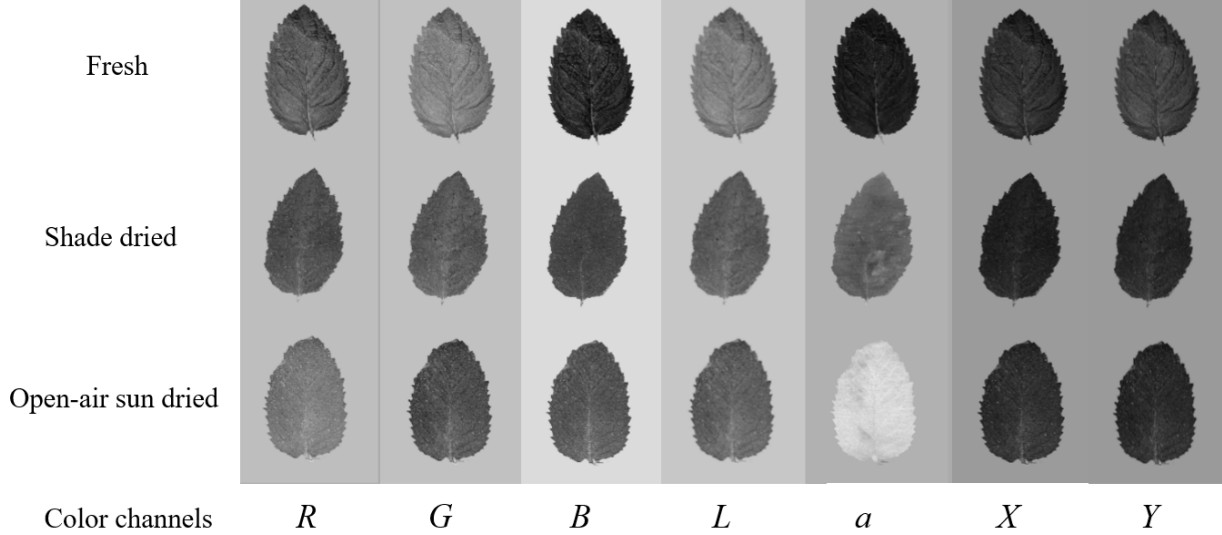

**Figure 3.** The images of the ventral side (upper surface) of fresh and dried mint leaves in selected color channels *R*, *G*, *B*, *L*, *a*, *X*, and *Y*.

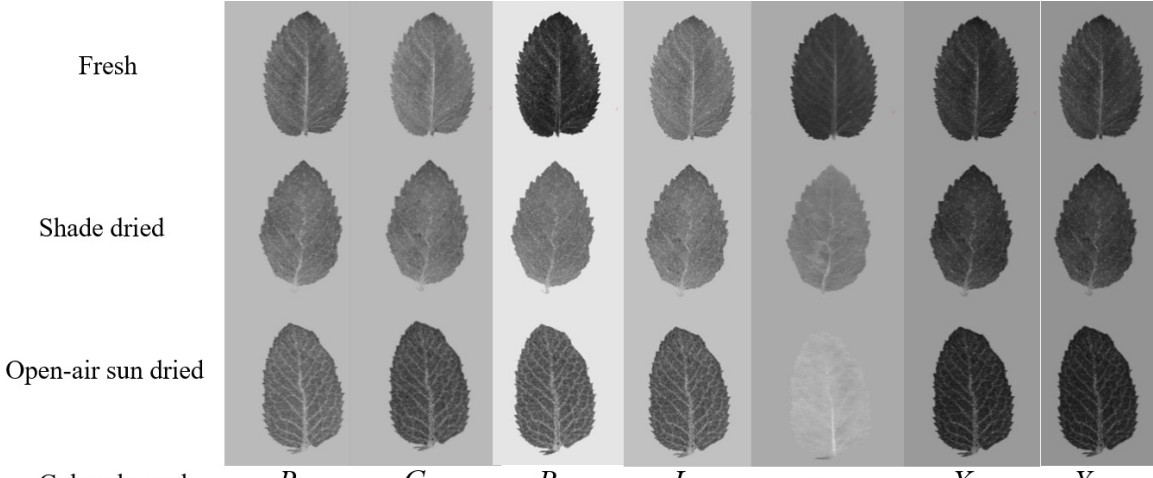

| | R | G | B | L | a | X | Y |

**Figure 4.** The images of the dorsal side (lower surface) of fresh and dried mint leaves in different color channels *R*, *G*, *B*, *L*, *a*, *X*, and *Y*.

*2.4. Classification of Fresh, Shade and Open-Air Sun-dried Mint Leaves*

The classification of leaf samples was performed using WEKA machine learning software (Machine Learning Group, University of Waikato) [21–23]. The influence of two drying techniques on the ventral and dorsal sides of mint leaves was compared. Models based on selected image features for individual color channels were built for distinguishing fresh vs. shade-dried mint leaves on the ventral side, fresh vs. open-air sun-dried mint leaves on the ventral side, fresh vs. shade-dried mint leaves on the dorsal side, and fresh vs. open-air sun-dried mint leaves on the dorsal side. First, the attribute (texture) selection was performed using the Best First algorithm with the Correlation-based Feature Selection (CFS) subset evaluator [24,25]. The attributes were selected separately from the datasets including texture parameters of images in each of color channels *R*, *G*, *B*, *L*, *a*, *b*, *X*, *Y*, and *Z* belonging to color spaces RGB, Lab, and XYZ. The classification analysis was performed using a test mode of 10-fold cross-validation. Various machine learning algorithms such as IBk, KStar and LWL from the group of Lazy, JRip and PART from Rules, Bayes Net and Naive Bayes from Bayes, Logistic, Multilayer Perceptron and RBF Classifier from Functions, Random Forest, J48 and LMT from Trees, and Filtered Classifier, Multi Class Classifier and Logit Boost from Meta were tested.

The methods chosen for classification are frequently used for machine learning. Any machine learning method used produces successful results if the extracted features strongly represent the target data. The physical changes caused by drying directly affect the pixel distribution in the image. For this, the texture-based extracted features are strong aids for distinguishing the drying method. In different spaces, the texture changes are also different. Some spaces show changes more clearly. For example, the green channel of the image is used in studies involving the segmentation of the retinal blood vessel, because in this channel the blood vessel is more vivid [26]. Therefore, examining the changes in texture on different channels may provide a more accurate distinction. In addition to the powerful features, the chosen classification algorithm also has a significant effect on success. Although strong features generally provide successful results, different learning methods provide different accuracy due to their methodology. In this context, extracting texture features from different channels and obtaining results with different classification algorithms makes the choice of method in this study powerful. The methodological steps of the study are shown in Figure 5. After all these steps, different result metrics and indicators showing the performance of the study are shared in the result section to show that all the experiments were done correctly. As a result, the confusion matrices with accuracies for each class, the average accuracies and values of Precision, Recall, F-Measure, and MCC (Matthews Correlation Coefficient) were determined [27–30].

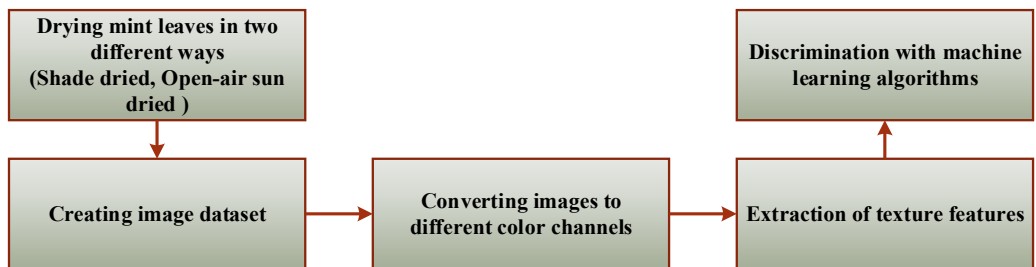

**Figure 5.** Algorithm steps applied to experimental study.

## 3. Results

The results of the classification of fresh mint leaves and leaves dried using two natural drying techniques were obtained. The effect of shade drying and open-air sun drying on the ventral and dorsal leaf sides was assessed. The distinguishing of samples was carried out using models including textures selected separately for each color channel from color spaces RGB, Lab, and XYZ. Models were developed using different machine learning algorithms. It was found that KStar from the group of Lazy, JRip from the group of Rules, and J48 from the group of Trees were the most effective. These algorithms provided the highest average accuracies of the classification of up to 100% for the selected datasets, which meant distinguishing the samples entirely correctly. Therefore, the results obtained by the KStar, JRip and J48 algorithms were chosen to be presented in this paper.

### 3.1. The Effect of Shade Drying and Open-Air Sun Drying on the Ventral Side (Upper Surface) of Mint Leaves

In the case of the ventral side (upper surface) of mint leaves, the classifications were carried out to compare the structure of fresh vs. shade-dried leaves and fresh vs. open-air sun-dried leaves. The models were built separately for color channels *R*, *G*, and *B* from color space RGB, color channels *L*, *a*, and *b* from color space Lab, and color channels *X*, *Y*, and *Z* from color space XYZ.

3.1.1. Classification of the Images of the Ventral Side of Fresh and Shade-dried Mint Leaves

For models built based on selected textures of images in color channels *R*, *G*, and *B* from color space RGB, fresh and shade-dried mint leaves were classified with an average accuracy reaching 100% only in the case of color channel *B* and KStar and J48 machine learning algorithms (Table 1). This meant that both fresh and shade-dried leaves were completely correctly classified. Also, the results of Precision, Recall, F-Measure, and MCC (Matthews Correlation Coefficient) equal to 1.000 were the most satisfactory. In the case of color channel *G*, all presented algorithms (KStar, JRip, J48) provided an average accuracy of 97.5%. However, models built using KStar and JRip classified fresh leaves with an accuracy of 95% and shade-dried leaves with an accuracy of 100%. In contrast, the model developed using J48 was characterized by an accuracy of 100% for fresh leaves and 95% for shade-dried leaves. The lowest accuracies of distinguishing fresh and shade-dried mint leaves were determined for models built based on textures selected from images in color channel *R*. For the KStar and J48 algorithms, average accuracy reached 92.5%. The model built using JRip provided the lowest average accuracy, at 90%, and values of Precision, Recall, and F-Measure of 0.900 and MCC of 0.800 for both fresh and shade-dried classes.

In the case of individual color channels from the color space Lab (Table 2), an average accuracy of 100% and values of Precision, Recall, F-Measure, and MCC of 1.000 were obtained for color channels *a* (KStar and J48) and *b* (KStar). Selected textures belonging to these channels allowed for building models that completely and correctly distinguished fresh and shade-dried mint leaves. For color channel *L*, the correctness of the classification of both samples was also high, up to 97.5% for KStar and J48. The model developed using the KStar algorithm correctly classified shade-dried leaves in 100% of cases and of fresh leaves in 95%, while the remaining 5% of cases belonging to the actual class of fresh

leaves were incorrectly classified as shade-dried ones. The application of J48 resulted in 100% correctness of the classification of fresh leaves and 95% for shade-dried samples.

**Table 1.** The classification of images of the ventral side of fresh and shade-dried mint leaves based on selected textures from color channels *R*, *G*, and *B* belonging to color space RGB.

| Algorithm | Predicted Class (%) Fresh | Shade-Dried | Actual Class | Average Accuracy (%) | Precision | Recall | F-Measure | MCC |
|---|---|---|---|---|---|---|---|---|
| | | | Color channel *R* | | | | | |
| lazy.KStar | 90 | 10 | Fresh | 92.5 | 0.947 | 0.900 | 0.923 | 0.851 |
| | 5 | 95 | Shade-dried | | 0.905 | 0.950 | 0.927 | 0.851 |
| rules.JRip | 90 | 10 | Fresh | 90 | 0.900 | 0.900 | 0.900 | 0.800 |
| | 10 | 90 | Shade-dried | | 0.900 | 0.900 | 0.900 | 0.800 |
| trees.J48 | 90 | 10 | Fresh | 92.5 | 0.947 | 0.900 | 0.923 | 0.851 |
| | 5 | 95 | Shade-dried | | 0.905 | 0.950 | 0.927 | 0.851 |
| | | | Color channel *G* | | | | | |
| lazy.KStar | 95 | 5 | Fresh | 97.5 | 1.000 | 0.950 | 0.974 | 0.951 |
| | 0 | 100 | Shade-dried | | 0.952 | 1.000 | 0.976 | 0.951 |
| rules.JRip | 95 | 5 | Fresh | 97.5 | 1.000 | 0.950 | 0.974 | 0.951 |
| | 0 | 100 | Shade-dried | | 0.952 | 1.000 | 0.976 | 0.951 |
| trees.J48 | 100 | 0 | Fresh | 97.5 | 0.952 | 1.000 | 0.976 | 0.951 |
| | 5 | 95 | Shade-dried | | 1.000 | 0.950 | 0.974 | 0.951 |
| | | | Color channel *B* | | | | | |
| lazy.KStar | 100 | 0 | Fresh | 100 | 1.000 | 1.000 | 1.000 | 1.000 |
| | 0 | 100 | Shade-dried | | 1.000 | 1.000 | 1.000 | 1.000 |
| rules.JRip | 95 | 5 | Fresh | 95 | 0.950 | 0.950 | 0.950 | 0.900 |
| | 5 | 95 | Shade-dried | | 0.950 | 0.950 | 0.950 | 0.900 |
| trees.J48 | 100 | 0 | Fresh | 100 | 1.000 | 1.000 | 1.000 | 1.000 |
| | 0 | 100 | Shade-dried | | 1.000 | 1.000 | 1.000 | 1.000 |

MCC—Matthews Correlation Coefficient.

**Table 2.** The results of the classification of fresh and shade-dried mint leaves imaged on the ventral side using models built based on selected textures from color channels *L*, *a*, and *b* from color space Lab.

| Algorithm | Predicted Class (%) Fresh | Shade-dried | Actual Class | Average Accuracy (%) | Precision | Recall | F-Measure | MCC |
|---|---|---|---|---|---|---|---|---|
| | | | Color channel *L* | | | | | |
| lazy.KStar | 95 | 5 | Fresh | 97.5 | 1.000 | 0.950 | 0.974 | 0.951 |
| | 0 | 100 | Shade-dried | | 0.952 | 1.000 | 0.976 | 0.951 |
| rules.JRip | 95 | 5 | Fresh | 95 | 0.950 | 0.950 | 0.950 | 0.900 |
| | 5 | 95 | Shade-dried | | 0.950 | 0.950 | 0.950 | 0.900 |
| trees.J48 | 100 | 0 | Fresh | 97.5 | 0.952 | 1.000 | 0.976 | 0.951 |
| | 5 | 95 | Shade-dried | | 1.000 | 0.950 | 0.974 | 0.951 |
| | | | Color channel *a* | | | | | |
| lazy.KStar | 100 | 0 | Fresh | 100 | 1.000 | 1.000 | 1.000 | 1.000 |
| | 0 | 100 | Shade-dried | | 1.000 | 1.000 | 1.000 | 1.000 |
| rules.JRip | 95 | 5 | Fresh | 97.5 | 1.000 | 0.950 | 0.974 | 0.951 |
| | 0 | 100 | Shade-dried | | 0.952 | 1.000 | 0.976 | 0.951 |
| trees.J48 | 100 | 0 | Fresh | 100 | 1.000 | 1.000 | 1.000 | 1.000 |
| | 0 | 100 | Shade-dried | | 1.000 | 1.000 | 1.000 | 1.000 |
| | | | Color channel *b* | | | | | |
| lazy.KStar | 100 | 0 | Fresh | 100 | 1.000 | 1.000 | 1.000 | 1.000 |
| | 0 | 100 | Shade-dried | | 1.000 | 1.000 | 1.000 | 1.000 |
| rules.JRip | 100 | 0 | Fresh | 97.5 | 0.952 | 1.000 | 0.976 | 0.951 |
| | 5 | 95 | Shade-dried | | 1.000 | 0.950 | 0.974 | 0.951 |
| trees.J48 | 100 | 0 | Fresh | 97.5 | 0.952 | 1.000 | 0.976 | 0.951 |
| | 5 | 95 | Shade-dried | | 1.000 | 0.950 | 0.974 | 0.951 |

MCC—Matthews Correlation Coefficient.

Fresh and shade-dried mint leaves imaged on the ventral side were also classified with a correctness reaching 100% and other classification performance metrics equal to 1.000 for one color channel (*Y*) from color space XYZ using the KStar algorithm (Table 3). Models built based on selected textures from images in color channels *X* and *Z* produced an average accuracy of up to 97.5%. In the case of color channel *X*, the model built using JRip revealed accuracies of 100% and 95% for fresh and shade-dried mint leaves, respectively. For J48, shade-dried leaves were correctly distinguished from fresh ones with an accuracy of 100%, fresh samples were correctly classified as fresh in 95% of cases, and the remaining 5% of cases were classified as shade-dried leaves. For the color channel *Z*, fresh leaves were correctly classified in 100% of cases and shade-dried samples in 95% of cases, for KStar and JRip.

**Table 3.** The performance metrics of distinguishing images of the ventral side of fresh and shade-dried mint leaves using selected textures from color channels *X*, *Y*, and *Z* from color space XYZ.

| Algorithm | Predicted Class (%) | | Actual Class | Average Accuracy (%) | Precision | Recall | F-Measure | MCC |
|---|---|---|---|---|---|---|---|---|
| | Fresh | Shade-Dried | | | | | | |
| | | | Color channel *X* | | | | | |
| lazy.KStar | 95 | 5 | Fresh | 95 | 0.950 | 0.950 | 0.950 | 0.900 |
| | 5 | 95 | Shade-dried | | 0.950 | 0.950 | 0.950 | 0.900 |
| rules.JRip | 100 | 0 | Fresh | 97.5 | 0.952 | 1.000 | 0.976 | 0.951 |
| | 5 | 95 | Shade-dried | | 1.000 | 0.950 | 0.974 | 0.951 |
| trees.J48 | 95 | 5 | Fresh | 97.5 | 1.000 | 0.950 | 0.974 | 0.951 |
| | 0 | 100 | Shade-dried | | 0.952 | 1.000 | 0.976 | 0.951 |
| | | | Color channel *Y* | | | | | |
| lazy.KStar | 100 | 0 | Fresh | 100 | 1.000 | 1.000 | 1.000 | 1.000 |
| | 0 | 100 | Shade-dried | | 1.000 | 1.000 | 1.000 | 1.000 |
| rules.JRip | 95 | 5 | Fresh | 97.5 | 1.000 | 0.950 | 0.974 | 0.951 |
| | 0 | 100 | Shade-dried | | 0.952 | 1.000 | 0.976 | 0.951 |
| trees.J48 | 95 | 5 | Fresh | 95 | 0.950 | 0.950 | 0.950 | 0.900 |
| | 5 | 95 | Shade-dried | | 0.950 | 0.950 | 0.950 | 0.900 |
| | | | Color channel *Z* | | | | | |
| lazy.KStar | 100 | 0 | Fresh | 97.5 | 0.952 | 1.000 | 0.976 | 0.951 |
| | 5 | 95 | Shade-dried | | 1.000 | 0.950 | 0.974 | 0.951 |
| rules.JRip | 100 | 0 | Fresh | 97.5 | 0.952 | 1.000 | 0.976 | 0.951 |
| | 5 | 95 | Shade-dried | | 1.000 | 0.950 | 0.974 | 0.951 |
| trees.J48 | 95 | 5 | Fresh | 95 | 0.950 | 0.950 | 0.950 | 0.900 |
| | 5 | 95 | Shade-dried | | 0.950 | 0.950 | 0.950 | 0.900 |

MCC—Matthews Correlation Coefficient.

### 3.1.2. Classification of the Images of the Ventral Side of Fresh and Open-Air Sun-dried Mint Leaves

In the next step of classification analysis, images of the ventral side of fresh and open-air sun-dried mint leaves were compared using models based on selected textures from individual color channels. Table 4 presents the results of distinguishing samples using selected textures extracted from images in color channels *R*, *G*, and *B*. The obtained results were very successful. An average accuracy of 100% and values of Precision, Recall, F-Measure, and MCC of 1.000 were found in the case of each color channel as channels *R* and *G* for J48, and channel *B* for all KStar, JRip, and J48 algorithms. It proved the high differentiation of fresh and open-air sun-dried mint leaves in terms of selected textures belonging to color space RGB and thus the great influence of open-air sun drying on the structure of the outer surface of the mint leaves.

Very high correctness of classification of fresh and open-air sun-dried mint leaves was also obtained for color channels *L*, *a*, and *b* from the color space Lab (Table 5). A completely correct classification with an average accuracy of 100% was observed in the case of models built based on selected texture parameters from images in color channels *a* and *b* for all algorithms (KStar, JRip, and J48). A slightly lower average accuracy reaching

97.5% was determined for a model built using selected textures from color channels *L* and the J48 algorithm.

**Table 4.** The distinguishing of fresh and open-air sun-dried mint leaves imaged on the ventral side based on models including selected textures from color channels *R*, *G*, and *B* belonging to color space RGB.

| Algorithm | Predicted Class (%) | | Actual Class | Average Accuracy (%) | Precision | Recall | F-Measure | MCC |
|---|---|---|---|---|---|---|---|---|
| | Fresh | Open-Air Sun-dried | | | | | | |
| | | | Color channel *R* | | | | | |
| lazy.KStar | 95 | 5 | Fresh | 95 | 0.950 | 0.950 | 0.950 | 0.900 |
| | 5 | 95 | Open-air sun-dried | | 0.950 | 0.950 | 0.950 | 0.900 |
| rules.JRip | 90 | 10 | Fresh | 92.5 | 0.947 | 0.900 | 0.923 | 0.851 |
| | 5 | 95 | Open-air sun-dried | | 0.905 | 0.950 | 0.927 | 0.851 |
| trees.J48 | 100 | 0 | Fresh | 100 | 1.000 | 1.000 | 1.000 | 1.000 |
| | 0 | 100 | Open-air sun-dried | | 1.000 | 1.000 | 1.000 | 1.000 |
| | | | Color channel *G* | | | | | |
| lazy.KStar | 100 | 0 | Fresh | 97.5 | 0.952 | 1.000 | 0.976 | 0.951 |
| | 5 | 95 | Open-air sun-dried | | 1.000 | 0.950 | 0.974 | 0.951 |
| rules.JRip | 90 | 10 | Fresh | 95 | 1.000 | 0.900 | 0.947 | 0.905 |
| | 0 | 100 | Open-air sun-dried | | 0.909 | 1.000 | 0.952 | 0.905 |
| trees.J48 | 100 | 0 | Fresh | 100 | 1.000 | 1.000 | 1.000 | 1.000 |
| | 0 | 100 | Open-air sun-dried | | 1.000 | 1.000 | 1.000 | 1.000 |
| | | | Color channel *B* | | | | | |
| lazy.KStar | 100 | 0 | Fresh | 100 | 1.000 | 1.000 | 1.000 | 1.000 |
| | 0 | 100 | Open-air sun-dried | | 1.000 | 1.000 | 1.000 | 1.000 |
| rules.JRip | 100 | 0 | Fresh | 100 | 1.000 | 1.000 | 1.000 | 1.000 |
| | 0 | 100 | Open-air sun-dried | | 1.000 | 1.000 | 1.000 | 1.000 |
| trees.J48 | 100 | 0 | Fresh | 100 | 1.000 | 1.000 | 1.000 | 1.000 |
| | 0 | 100 | Open-air sun-dried | | 1.000 | 1.000 | 1.000 | 1.000 |

MCC—Matthews Correlation Coefficient.

**Table 5.** The classification of images of the ventral side of fresh and open-air sun-dried mint leaves using models including selected textures from color channels *L*, *a*, and *b* from color space Lab.

| Algorithm | Predicted Class (%) | | Actual Class | Average Accuracy (%) | Precision | Recall | F-Measure | MCC |
|---|---|---|---|---|---|---|---|---|
| | Fresh | Open-Air Sun-dried | | | | | | |
| | | | Color channel *L* | | | | | |
| lazy.KStar | 95 | 5 | Fresh | 95 | 0.950 | 0.950 | 0.950 | 0.900 |
| | 5 | 95 | Open-air sun-dried | | 0.950 | 0.950 | 0.950 | 0.900 |
| rules.JRip | 95 | 5 | Fresh | 95 | 0.950 | 0.950 | 0.950 | 0.900 |
| | 5 | 95 | Open-air sun-dried | | 0.950 | 0.950 | 0.950 | 0.900 |
| trees.J48 | 100 | 0 | Fresh | 97.5 | 0.952 | 1.000 | 0.976 | 0.951 |
| | 5 | 95 | Open-air sun-dried | | 1.000 | 0.950 | 0.974 | 0.951 |
| | | | Color channel *a* | | | | | |
| lazy.KStar | 100 | 0 | Fresh | 100 | 1.000 | 1.000 | 1.000 | 1.000 |
| | 0 | 100 | Open-air sun-dried | | 1.000 | 1.000 | 1.000 | 1.000 |
| rules.JRip | 100 | 0 | Fresh | 100 | 1.000 | 1.000 | 1.000 | 1.000 |
| | 0 | 100 | Open-air sun-dried | | 1.000 | 1.000 | 1.000 | 1.000 |
| trees.J48 | 100 | 0 | Fresh | 100 | 1.000 | 1.000 | 1.000 | 1.000 |
| | 0 | 100 | Open-air sun-dried | | 1.000 | 1.000 | 1.000 | 1.000 |
| | | | Color channel *b* | | | | | |
| lazy.KStar | 100 | 0 | Fresh | 100 | 1.000 | 1.000 | 1.000 | 1.000 |
| | 0 | 100 | Open-air sun-dried | | 1.000 | 1.000 | 1.000 | 1.000 |
| rules.JRip | 100 | 0 | Fresh | 100 | 1.000 | 1.000 | 1.000 | 1.000 |
| | 0 | 100 | Open-air sun-dried | | 1.000 | 1.000 | 1.000 | 1.000 |
| trees.J48 | 100 | 0 | Fresh | 100 | 1.000 | 1.000 | 1.000 | 1.000 |
| | 0 | 100 | Open-air sun-dried | | 1.000 | 1.000 | 1.000 | 1.000 |

MCC—Matthews Correlation Coefficient.

The models built using selected textures from color channels belonging to color space XYZ of images of the ventral side of leaves produced the lowest results for distinguishing fresh and open-air sun-dried mint classes (Table 6). Samples were completely correctly classified only in the case of models developed using the KStar algorithm for selected textures from color channels $X$ and $Y$. For other machine learning algorithms of JRip and J48, an average accuracy of 95% was obtained for color channel $X$ and 97.5% for color channel $Y$. An average accuracy of the classification of fresh and open-air sun-dried mint leaves reached 97.5% for a model including selected textures from color channel $Z$ developed using J48.

**Table 6.** The classification performance metrics for fresh vs. open-air sun-dried mint leaves imaged on the ventral side based on models including selected textures from color channels $X$, $Y$, and $Z$ from color space XYZ.

| Algorithm | Predicted Class (%) | | Actual Class | Average Accuracy (%) | Precision | Recall | F-Measure | MCC |
|---|---|---|---|---|---|---|---|---|
| | Fresh | Open-Air Sun-dried | | | | | | |
| | | | Color channel $X$ | | | | | |
| lazy.KStar | 100 | 0 | Fresh | 100 | 1.000 | 1.000 | 1.000 | 1.000 |
| | 0 | 100 | Open-air sun-dried | | 1.000 | 1.000 | 1.000 | 1.000 |
| rules.JRip | 95 | 5 | Fresh | 95 | 0.950 | 0.950 | 0.950 | 0.900 |
| | 5 | 95 | Open-air sun-dried | | 0.950 | 0.950 | 0.950 | 0.900 |
| trees.J48 | 95 | 5 | Fresh | 95 | 0.950 | 0.950 | 0.950 | 0.900 |
| | 5 | 95 | Open-air sun-dried | | 0.950 | 0.950 | 0.950 | 0.900 |
| | | | Color channel $Y$ | | | | | |
| lazy.KStar | 100 | 0 | Fresh | 100 | 1.000 | 1.000 | 1.000 | 1.000 |
| | 0 | 100 | Open-air sun-dried | | 1.000 | 1.000 | 1.000 | 1.000 |
| rules.JRip | 95 | 5 | Fresh | 97.5 | 1.000 | 0.950 | 0.974 | 0.951 |
| | 0 | 100 | Open-air sun-dried | | 0.952 | 1.000 | 0.976 | 0.951 |
| trees.J48 | 95 | 5 | Fresh | 97.5 | 1.000 | 0.950 | 0.974 | 0.951 |
| | 0 | 100 | Open-air sun-dried | | 0.952 | 1.000 | 0.976 | 0.951 |
| | | | Color channel $Z$ | | | | | |
| lazy.KStar | 95 | 5 | Fresh | 92.5 | 0.905 | 0.950 | 0.927 | 0.851 |
| | 10 | 90 | Open-air sun-dried | | 0.947 | 0.900 | 0.923 | 0.851 |
| rules.JRip | 90 | 10 | Fresh | 92.5 | 0.947 | 0.900 | 0.923 | 0.851 |
| | 5 | 95 | Open-air sun-dried | | 0.905 | 0.950 | 0.927 | 0.851 |
| trees.J48 | 100 | 0 | Fresh | 97.5 | 0.952 | 1.000 | 0.976 | 0.951 |
| | 5 | 95 | Open-air sun-dried | | 1.000 | 0.950 | 0.974 | 0.951 |

MCC—Matthews Correlation Coefficient.

*3.2. The Effect of Shade Drying and Open-Air Sun Drying on the Dorsal Side (Lower Surface) of Mint Leaves*

To assess the effect of different natural drying techniques on the structure of the dorsal side of mint leaves, various classification models were developed. In the case of both the classification of fresh vs. shade-dried mint leaves and fresh vs. open-air sun-dried leaves, models included image textures selected separately for individual color channels $R$, $G$, $B$, $L$, $a$, $b$, $X$, $Y$, and $Z$.

3.2.1. Classification of the Images of the Dorsal Side of Fresh and Shade-dried Mint Leaves

In the case of color channels $R$, $G$, and $B$ from color space RGB, the images of the dorsal side of shade-dried mint leaves were distinguished from the fresh ones with an average accuracy of 75% for a model built based on selected textures from color channel $G$ using the KStar algorithm to 100% in the case of color channel $B$ and KStar (Table 7). For the model producing 75% correctness, fresh leaves were classified with an accuracy of 70% and shade-dried samples with an accuracy of 80%. The values of Precision, Recall, and F-Measure were in the range of 0.700–0.800, and MCC was equal to 0.503 for both classes. These results are relatively low compared with the 100% accuracy and values of other metrics of 1.000 for one model including selected texture parameters extracted from images in color channel $B$.

For color channel *G*, average accuracy reached 82.5% (75% for fresh leaves and 90% for shade-dried leaves) for the J48 algorithm. For models built for selected textures from images in color channel *R*, average accuracy ranged from 77.5% (KStar) to 90% (JRip).

**Table 7.** The performance metrics of the classification of the dorsal side of fresh and shade-dried mint leaves based on selected textures of images in color channels *R*, *G*, and *B* from color space RGB.

| Algorithm | Predicted Class (%) Fresh | Shade-Dried | Actual Class | Average Accuracy (%) | Precision | Recall | F-Measure | MCC |
|---|---|---|---|---|---|---|---|---|
| | | | Color channel *R* | | | | | |
| lazy.KStar | 75 | 25 | Fresh | 77.5 | 0.789 | 0.750 | 0.769 | 0.551 |
| | 20 | 80 | Shade-dried | | 0.762 | 0.800 | 0.780 | 0.551 |
| rules.JRip | 85 | 15 | Fresh | 90 | 0.944 | 0.850 | 0.895 | 0.804 |
| | 5 | 95 | Shade-dried | | 0.864 | 0.950 | 0.905 | 0.804 |
| trees.J48 | 85 | 15 | Fresh | 87.5 | 0.895 | 0.850 | 0.872 | 0.751 |
| | 10 | 90 | Shade-dried | | 0.857 | 0.900 | 0.878 | 0.751 |
| | | | Color channel *G* | | | | | |
| lazy.KStar | 70 | 30 | Fresh | 75 | 0.778 | 0.700 | 0.737 | 0.503 |
| | 20 | 80 | Shade-dried | | 0.727 | 0.800 | 0.762 | 0.503 |
| rules.JRip | 85 | 15 | Fresh | 77.5 | 0.739 | 0.850 | 0.791 | 0.556 |
| | 30 | 70 | Shade-dried | | 0.824 | 0.700 | 0.757 | 0.556 |
| trees.J48 | 75 | 25 | Fresh | 82.5 | 0.882 | 0.750 | 0.811 | 0.657 |
| | 10 | 90 | Shade-dried | | 0.783 | 0.900 | 0.837 | 0.657 |
| | | | Color channel *B* | | | | | |
| lazy.KStar | 100 | 0 | Fresh | 100 | 1.000 | 1.000 | 1.000 | 1.000 |
| | 0 | 100 | Shade-dried | | 1.000 | 1.000 | 1.000 | 1.000 |
| rules.JRip | 100 | 0 | Fresh | 97.5 | 0.952 | 1.000 | 0.976 | 0.951 |
| | 5 | 95 | Shade-dried | | 1.000 | 0.950 | 0.974 | 0.951 |
| trees.J48 | 95 | 5 | Fresh | 97.5 | 1.000 | 0.950 | 0.974 | 0.951 |
| | 0 | 100 | Shade-dried | | 0.952 | 1.000 | 0.976 | 0.951 |

MCC—Matthews Correlation Coefficient.

The higher accuracies of the classification of images of the dorsal side of fresh and shade-dried mint leaves were obtained for the models built based on textures selected separately for individual color channels of Lab space (Table 8). It indicated that the differences in the image textures were more noticeable. All applied machine learning algorithms provided 100% accuracy for textures from images in color channel *a*. Slightly lower average accuracies reaching 97.5% (KStar, J48) and 95% (J48) were found in the case of the color channels *b* and *L*, respectively.

**Table 8.** The classification of the dorsal side of fresh and shade-dried mint leaves using models including selected image textures from color channels *L*, *a*, and *b*.

| Algorithm | Predicted Class (%) Fresh | Shade-Dried | Actual Class | Average Accuracy (%) | Precision | Recall | F-Measure | MCC |
|---|---|---|---|---|---|---|---|---|
| | | | Color channel *L* | | | | | |
| lazy.KStar | 70 | 30 | Fresh | 77.5 | 0.824 | 0.700 | 0.757 | 0.556 |
| | 15 | 85 | Shade-dried | | 0.739 | 0.850 | 0.791 | 0.556 |
| rules.JRip | 90 | 10 | Fresh | 92.5 | 0.947 | 0.900 | 0.923 | 0.851 |
| | 5 | 95 | Shade-dried | | 0.905 | 0.950 | 0.927 | 0.851 |
| trees.J48 | 100 | 0 | Fresh | 95 | 0.909 | 1.000 | 0.952 | 0.905 |
| | 10 | 90 | Shade-dried | | 1.000 | 0.900 | 0.947 | 0.905 |
| | | | Color channel *a* | | | | | |
| lazy.KStar | 100 | 0 | Fresh | 100 | 1.000 | 1.000 | 1.000 | 1.000 |
| | 0 | 100 | Shade-dried | | 1.000 | 1.000 | 1.000 | 1.000 |
| rules.JRip | 100 | 0 | Fresh | 100 | 1.000 | 1.000 | 1.000 | 1.000 |
| | 0 | 100 | Shade-dried | | 1.000 | 1.000 | 1.000 | 1.000 |
| trees.J48 | 100 | 0 | Fresh | 100 | 1.000 | 1.000 | 1.000 | 1.000 |
| | 0 | 100 | Shade-dried | | 1.000 | 1.000 | 1.000 | 1.000 |

**Table 8.** *Cont.*

| Algorithm | Predicted Class (%) Fresh | Shade-Dried | Actual Class | Average Accuracy (%) | Precision | Recall | F-Measure | MCC |
|---|---|---|---|---|---|---|---|---|
| | | | Color channel *b* | | | | | |
| lazy.KStar | 100 | 0 | Fresh | 97.5 | 0.952 | 1.000 | 0.976 | 0.951 |
| | 5 | 95 | Shade-dried | | 1.000 | 0.950 | 0.974 | 0.951 |
| rules.JRip | 95 | 5 | Fresh | 95 | 0.950 | 0.950 | 0.950 | 0.900 |
| | 5 | 95 | Shade-dried | | 0.950 | 0.950 | 0.950 | 0.900 |
| trees.J48 | 100 | 0 | Fresh | 97.5 | 0.952 | 1.000 | 0.976 | 0.951 |
| | 5 | 95 | Shade-dried | | 1.000 | 0.950 | 0.974 | 0.951 |

MCC—Matthews Correlation Coefficient.

In the case of models including selected image textures from color channels *X*, *Y*, and *Z*, a completely correct classification was not observed (Table 9). The differentiation of the structure of the dorsal side of fresh and shade-dried mint leaves allowed for distinguishing both samples at an average accuracy of up to 97.5% (100% for fresh and 95% for dried leaves) for color channel *X* and J48 as well as color channel *Z* and KStar and J48. An average accuracy of 95% was determined for models developed based on selected textures of images in color channel *Y* using all algorithms.

**Table 9.** The results of the classification of fresh and shade-dried mint leaves imaged on the dorsal side using models built based on selected texture parameters from color channels *X*, *Y*, and *Z* of images.

| Algorithm | Predicted Class (%) Fresh | Shade-Dried | Actual Class | Average Accuracy (%) | Precision | Recall | F-Measure | MCC |
|---|---|---|---|---|---|---|---|---|
| | | | Color channel *X* | | | | | |
| lazy.KStar | 90 | 10 | Fresh | 92.5 | 0.947 | 0.900 | 0.923 | 0.851 |
| | 5 | 95 | Shade-dried | | 0.905 | 0.950 | 0.927 | 0.851 |
| rules.JRip | 95 | 5 | Fresh | 95 | 0.950 | 0.950 | 0.950 | 0.900 |
| | 5 | 95 | Shade-dried | | 0.950 | 0.950 | 0.950 | 0.900 |
| trees.J48 | 100 | 0 | Fresh | 97.5 | 0.952 | 1.000 | 0.976 | 0.951 |
| | 5 | 95 | Shade-dried | | 1.000 | 0.950 | 0.974 | 0.951 |
| | | | Color channel *Y* | | | | | |
| lazy.KStar | 95 | 5 | Fresh | 95 | 0.950 | 0.950 | 0.950 | 0.900 |
| | 5 | 95 | Shade-dried | | 0.950 | 0.950 | 0.950 | 0.900 |
| rules.JRip | 95 | 5 | Fresh | 95 | 0.950 | 0.950 | 0.950 | 0.900 |
| | 5 | 95 | Shade-dried | | 0.950 | 0.950 | 0.950 | 0.900 |
| trees.J48 | 95 | 5 | Fresh | 95 | 0.950 | 0.950 | 0.950 | 0.900 |
| | 5 | 95 | Shade-dried | | 0.950 | 0.950 | 0.950 | 0.900 |
| | | | Color channel *Z* | | | | | |
| lazy.KStar | 100 | 0 | Fresh | 97.5 | 0.952 | 1.000 | 0.976 | 0.951 |
| | 5 | 95 | Shade-dried | | 1.000 | 0.950 | 0.974 | 0.951 |
| rules.JRip | 95 | 5 | Fresh | 95 | 0.950 | 0.950 | 0.950 | 0.900 |
| | 5 | 95 | Shade-dried | | 0.950 | 0.950 | 0.950 | 0.900 |
| trees.J48 | 100 | 0 | Fresh | 97.5 | 0.952 | 1.000 | 0.976 | 0.951 |
| | 5 | 95 | Shade-dried | | 1.000 | 0.950 | 0.974 | 0.951 |

MCC—Matthews Correlation Coefficient.

### 3.2.2. Classification of the Images of the Dorsal Side of Fresh and Open-Air Sun-dried Mint Leaves

Very successful classifications were performed for the fresh and open-air sun-dried mint leaves using models built based on selected textures of the dorsal leaf side images converted to color channels *R*, *G*, and *B* (Table 10). Average accuracy reached 97.5% for models developed using the KStar, JRip, and J48 algorithms based on features of images in color channel *B*. All the other models produced an average accuracy of 92.5% in the case of color channels *R* and *G*.

**Table 10.** The differentiation of fresh and open-air sun-dried mint leaves in terms of selected image textures of the dorsal leaf side based on models built for color channels *R*, *G*, and *B* from color space RGB.

| Algorithm | Predicted Class (%) | | Actual Class | Average Accuracy (%) | Precision | Recall | F-Measure | MCC |
|---|---|---|---|---|---|---|---|---|
| | Fresh | Open-Air Sun-dried | | | | | | |
| | | | Color channel *R* | | | | | |
| lazy.KStar | 95 | 5 | Fresh | 92.5 | 0.905 | 0.950 | 0.927 | 0.851 |
| | 10 | 90 | Open-air sun-dried | | 0.947 | 0.900 | 0.923 | 0.851 |
| rules.JRip | 100 | 0 | Fresh | 92.5 | 0.870 | 1.000 | 0.930 | 0.860 |
| | 15 | 85 | Open-air sun-dried | | 1.000 | 0.850 | 0.919 | 0.860 |
| trees.J48 | 100 | 0 | Fresh | 92.5 | 0.870 | 1.000 | 0.930 | 0.860 |
| | 15 | 85 | Open-air sun-dried | | 1.000 | 0.850 | 0.919 | 0.860 |
| | | | Color channel *G* | | | | | |
| lazy.KStar | 95 | 5 | Fresh | 92.5 | 0.905 | 0.950 | 0.927 | 0.851 |
| | 10 | 90 | Open-air sun-dried | | 0.947 | 0.900 | 0.923 | 0.851 |
| rules.JRip | 100 | 0 | Fresh | 92.5 | 0.870 | 1.000 | 0.930 | 0.860 |
| | 15 | 85 | Open-air sun-dried | | 1.000 | 0.850 | 0.919 | 0.860 |
| trees.J48 | 100 | 0 | Fresh | 92.5 | 0.870 | 1.000 | 0.930 | 0.860 |
| | 15 | 85 | Open-air sun-dried | | 1.000 | 0.850 | 0.919 | 0.860 |
| | | | Color channel *B* | | | | | |
| lazy.KStar | 100 | 0 | Fresh | 97.5 | 0.952 | 1.000 | 0.976 | 0.951 |
| | 5 | 95 | Open-air sun-dried | | 1.000 | 0.950 | 0.974 | 0.951 |
| rules.JRip | 95 | 5 | Fresh | 97.5 | 1.000 | 0.950 | 0.974 | 0.951 |
| | 0 | 100 | Open-air sun-dried | | 0.952 | 1.000 | 0.976 | 0.951 |
| trees.J48 | 95 | 5 | Fresh | 97.5 | 1.000 | 0.950 | 0.974 | 0.951 |
| | 0 | 100 | Open-air sun-dried | | 0.952 | 1.000 | 0.976 | 0.951 |

MCC—Matthews Correlation Coefficient.

In the case of the color space Lab, images of the dorsal side of fresh and open-air sun-dried mint leaves turned out to be completely different in terms of selected textures from color channels *a* (KStar, JRip, J48) and *b* (J48) (Table 11). The accuracies were equal to 100% and the values of Precision, Recall, F-Measure, and MCC were 1.000 for both classes. In the case of the color channel *L*, the correctness of classification was slightly lower, reaching 97.5% for the model built using the JRip machine learning algorithm.

**Table 11.** The distinguishing of fresh and open-air sun-dried mint leaves imaged on the dorsal side using models developed based on selected textures from color channels *L*, *a*, and *b* from color space Lab.

| Algorithm | Predicted Class (%) | | Actual Class | Average Accuracy (%) | Precision | Recall | F-Measure | MCC |
|---|---|---|---|---|---|---|---|---|
| | Fresh | Open-Air Sun-dried | | | | | | |
| | | | Color channel *L* | | | | | |
| lazy.KStar | 95 | 5 | Fresh | 90 | 0.864 | 0.950 | 0.905 | 0.804 |
| | 15 | 85 | Open-air sun-dried | | 0.944 | 0.850 | 0.895 | 0.804 |
| rules.JRip | 100 | 0 | Fresh | 97.5 | 0.952 | 1.000 | 0.976 | 0.951 |
| | 5 | 95 | Open-air sun-dried | | 1.000 | 0.950 | 0.974 | 0.951 |
| trees.J48 | 100 | 0 | Fresh | 95 | 0.909 | 1.000 | 0.952 | 0.905 |
| | 10 | 90 | Open-air sun-dried | | 1.000 | 0.900 | 0.947 | 0.905 |
| | | | Color channel *a* | | | | | |
| lazy.KStar | 100 | 0 | Fresh | 100 | 1.000 | 1.000 | 1.000 | 1.000 |
| | 0 | 100 | Open-air sun-dried | | 1.000 | 1.000 | 1.000 | 1.000 |
| rules.JRip | 100 | 0 | Fresh | 100 | 1.000 | 1.000 | 1.000 | 1.000 |
| | 0 | 100 | Open-air sun-dried | | 1.000 | 1.000 | 1.000 | 1.000 |
| trees.J48 | 100 | 0 | Fresh | 100 | 1.000 | 1.000 | 1.000 | 1.000 |
| | 0 | 100 | Open-air sun-dried | | 1.000 | 1.000 | 1.000 | 1.000 |

**Table 11.** *Cont.*

| Algorithm | Predicted Class (%) | | Actual Class | Average Accuracy (%) | Precision | Recall | F-Measure | MCC |
|---|---|---|---|---|---|---|---|---|
| | Fresh | Open-Air Sun-dried | | | | | | |
| | | | Color channel *b* | | | | | |
| lazy.KStar | 100 | 0 | Fresh | 97.5 | 0.952 | 1.000 | 0.976 | 0.951 |
| | 5 | 95 | Open-air sun-dried | | 1.000 | 0.950 | 0.974 | 0.951 |
| rules.JRip | 100 | 0 | Fresh | 97.5 | 0.952 | 1.000 | 0.976 | 0.951 |
| | 5 | 95 | Open-air sun-dried | | 1.000 | 0.950 | 0.974 | 0.951 |
| trees.J48 | 100 | 0 | Fresh | 100 | 1.000 | 1.000 | 1.000 | 1.000 |
| | 0 | 100 | Open-air sun-dried | | 1.000 | 1.000 | 1.000 | 1.000 |

MCC—Matthews Correlation Coefficient.

The classification of fresh and open-air sun-dried mint leaves based on image textures of the dorsal leaf side revealed 100% correctness in the case of color channel *Z* and the J48 algorithm (Table 12). For channel *Z*, other algorithms (KStar, JRip) also produced a high average accuracy of 97.5%. Furthermore, an average accuracy of 97.5% was found for models built using KStar and J48 based on selected textures of images in color channel *Y*. In the case of individual color channels from color space XYZ, the lowest correctness of distinguishing fresh and open-air sun-dried mint leaves based on image textures of the dorsal leaf side was determined for color channel *X*. Average accuracy reached 95% for a model developed using J48.

**Table 12.** The performance metrics of distinguishing fresh and open-air sun-dried mint leaves using textures from images of the dorsal leaf side in color channels *X*, *Y*, and *Z* from color space XYZ.

| Algorithm | Predicted Class (%) | | Actual Class | Average Accuracy (%) | Precision | Recall | F-Measure | MCC |
|---|---|---|---|---|---|---|---|---|
| | Fresh | Open-Air Sun-dried | | | | | | |
| | | | Color channel *X* | | | | | |
| lazy.KStar | 95 | 5 | Fresh | 92.5 | 0.905 | 0.950 | 0.927 | 0.851 |
| | 10 | 90 | Open-air sun-dried | | 0.947 | 0.900 | 0.923 | 0.851 |
| rules.JRip | 95 | 5 | Fresh | 90 | 0.864 | 0.950 | 0.905 | 0.804 |
| | 15 | 85 | Open-air sun-dried | | 0.944 | 0.850 | 0.895 | 0.804 |
| trees.J48 | 100 | 0 | Fresh | 95 | 0.909 | 1.000 | 0.952 | 0.905 |
| | 10 | 90 | Open-air sun-dried | | 1.000 | 0.900 | 0.947 | 0.905 |
| | | | Color channel *Y* | | | | | |
| lazy.KStar | 100 | 0 | Fresh | 97.5 | 0.952 | 1.000 | 0.976 | 0.951 |
| | 5 | 95 | Open-air sun-dried | | 1.000 | 0.950 | 0.974 | 0.951 |
| rules.JRip | 90 | 10 | Fresh | 92.5 | 0.947 | 0.900 | 0.923 | 0.851 |
| | 5 | 95 | Open-air sun-dried | | 0.905 | 0.950 | 0.927 | 0.851 |
| trees.J48 | 100 | 0 | Fresh | 97.5 | 0.952 | 1.000 | 0.976 | 0.951 |
| | 5 | 95 | Open-air sun-dried | | 1.000 | 0.950 | 0.974 | 0.951 |
| | | | Color channel *Z* | | | | | |
| lazy.KStar | 100 | 0 | Fresh | 97.5 | 0.952 | 1.000 | 0.976 | 0.951 |
| | 5 | 95 | Open-air sun-dried | | 1.000 | 0.950 | 0.974 | 0.951 |
| rules.JRip | 100 | 0 | Fresh | 97.5 | 0.952 | 1.000 | 0.976 | 0.951 |
| | 5 | 95 | Open-air sun-dried | | 1.000 | 0.950 | 0.974 | 0.951 |
| trees.J48 | 100 | 0 | Fresh | 100 | 1.000 | 1.000 | 1.000 | 1.000 |
| | 0 | 100 | Open-air sun-dried | | 1.000 | 1.000 | 1.000 | 1.000 |

MCC—Matthews Correlation Coefficient.

Texture features from images in color channels *R*, *G*, *B*, *L*, *a*, *b*, *X*, *Y*, and *Z* and artificial intelligence involving models built by traditional machine learning algorithms proved to be useful for distinguishing fresh and shade drying or open-air sun-dried mint leaves. Machine learning-based algorithms are considered an innovative approach to advancing food drying technology. Machine learning algorithms have a superior capacity to predict specific trends and patterns using large volumes of data. Machine learning-based strategies are used, among others, for drying modeling to determine the optimal drying condition to improve product quality and reduce energy consumption. Machine learning models

can also be used for modeling to predict food properties during drying [31]. Machine learning found application for the monitoring of different drying processes employed in industry, for example, of convective drying, osmotic-convective drying, microwave drying, infrared drying, microwave- and infrared-assisted drying, fluidized bed drying, spouted bed drying, spray drying, rotary drying, deep bed drying, renewable drying, and freeze drying [13,32]. In the case of mint, machine learning was also used to investigate drying behavior and assess drying kinetics [33]. Additionally, machine learning algorithms were successfully applied to discriminate between different mint samples [34]. Our research has set new directions for the application of machine learning in the drying and discrimination of mint. The research carried out with the use of traditional machine learning models may be expanded in the future to include deep learning to distinguish fresh and dried mint leaves.

### 4. Conclusions

This study aimed to analyze the changes caused by two different drying methods on mint leaves. Because these changes cause significant changes in the nutrient content of the leaf, it is important to provide drying with the least loss and to develop additional techniques to do so. In order to clearly observe the effect of drying, the detectability of these changes with computer vision and machine learning algorithms is adopted as a hypothesis. The most obvious changes as a result of drying occur in leaf textures, and therefore texture analysis is performed for computer vision. In experimental studies, different image spaces and different machine learning techniques are used to prove that the proposed method is scientifically sound.

The combination of digital color imaging and machine learning proved to be an effective approach to the evaluation of the effect of natural drying techniques on mint leaf structure. Models built based on selected textures of images in color channels $R$, $G$, $B$, $L$, $a$, $b$, $X$, $Y$, and $Z$ were useful for monitoring the changes in the ventral side (upper surface) and dorsal side (lower surface) of mint leaves caused by shade drying and open-air sun drying. Fresh and dried leaves were distinguished with an accuracy of up to 100% using selected machine learning algorithms from groups of Lazy, Rules, and Trees. The obtained results were promising, and the developed procedure involving image features and machine learning could be used to evaluate the quality of mint leaves dried by other techniques. When the accuracy rates obtained as a result of experimental studies are examined, it is seen that the discrimination accuracy in all steps is generally above 90%. Compliance and consistency of all results indicate the accuracy and robustness of the results. Besides traditional machine learning, deep learning as well could be applied in further studies for the classification of fresh and dried leaves. For the deep learning application, data with more samples will be created in the future, and experiments will be carried out with different deep learning models on this data. Different drying techniques will also be included in the dataset.

**Author Contributions:** Conceptualization, E.R.; methodology, E.R.; software, E.R.; validation, E.R., K.S. and M.F.A.; formal analysis, E.R.; investigation, E.R.; resources, E.R.; data curation, E.R.; writing—original draft preparation, E.R.; writing—review and editing, E.R., K.S. and M.F.A.; visualization, E.R.; supervision, E.R. All authors have read and agreed to the published version of the manuscript.

**Funding:** This research received no external funding.

**Institutional Review Board Statement:** Not applicable.

**Informed Consent Statement:** Not applicable.

**Data Availability Statement:** The data presented in this study are available on request from the corresponding author.

**Conflicts of Interest:** The authors declare no conflict of interest.

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
