# Peer review of "The Use of Digital Color Imaging and Machine Learning for the Evaluation of the Effects of Shade Drying and Open-Air Sun Drying on Mint Leaf Quality"

_applsci, doi:10.3390/app13010206_

Round 1

Reviewer 1 Report

It can be recommended to improve the experiment describing by mean of such additional points:

1)      it would be necessary to prove that the chosen mathematical methods for the experiment are better than others, for example, methods based on fuzzy logic rules or others

2)      to include the formal statement of the problem and the algorithm for it solving.

Author Response

Responses to Reviewer 1:

Comment 1. it would be necessary to prove that the chosen mathematical methods for the experiment are better than others, for example, methods based on fuzzy logic rules or others

Response 1.   Thank you for the comment. More details on method selection have been added. Many algorithms have been tested. As the selected methods ensured 100% correctness for the selected datasets, there was no need to look for other methods.

Various machine learning algorithms such as IBk, KStar and LWL from the group of Lazy, JRip and PART from Rules, Bayes Net and Naive Bayes from Bayes, Logistic, Multilayer Perceptron and RBF Classifier from Functions, Random Forest, J48 and LMT from Trees, and Filtered Classifier, Multi Class Classifier and Logit Boost from Meta were tested.

It was found that KStar from the group of Lazy, JRip from the group of Rules, and J48 from the group of Trees were the most effective. These algorithms provided the highest average accuracies of the classification of up to 100% for the selected datasets. It meant completely correct distinguishing the samples. Therefore, the results obtained by the KStar, JRip and J48 algorithms were chosen to be presented in this paper.

 Comment 2. to include the formal statement of the problem and the algorithm for it solving.

The problem addressed in this study is to reveal the textural changes caused by the drying techniques used to preserve the mint leaf for a long time, with technological solutions. The proposed solution for this is to combine the power of computer vision and artificial intelligence, which are used in many different agricultural applications today [16,17]. In this way, discrimination systems with non-destructive, non-biased, high-accuracy and autonomous capabilities can be developed. The proposed solution also does not depend on a single artificial intelligence algorithm or a single image space. The robustness of the proposed approach is proven with different machine learning methods and different image spaces applied experimentally.

 Response 2. Thanks a lot for your valuable opinion. The revision is as follows. Please also see the revised manuscript.

Reviewer 2 Report

The aim of this study is to compare the effects of shade drying and sun-drying in the open air on the ventral (upper surface) and dorsal (lower surface) sides of mint leaves. The article presents the results of leaf classification using traditional classification methods. The topic of the article is quite interesting, there is a practical problem that needs to be solved using machine learning methods. However, it is not clear what is the novelty of the article, as classical methods are used to solve the specific problem. The current state of the work is unacceptable, and major changes are needed to improve its quality.

The novelty of this study is justified by the fact that The evaluation of the changes in mint leaves caused by two natural drying techniques was performed using models based on selected texture parameters extracted from the digital color images converted to color channels R, G, B, L, a, b, X, Y, and Z built using machine learning algorithms from groups of Lazy, Rules, and Trees.

However, this statement is not entirely clear, and an insufficient argument for the novelty of this study. More emphasis should be placed on the novelty of the proposed method (research or solution) and its impact on the field (what new advance or contribution the paper represents). There are no new methods or solutions to specific problems in the article, only data analysis and experimental results are provided.

It is not substantiated that the manuscript is scientifically sound, the experimental design to test the hypothesis is chosen correctly, and all experiments performed are done correctly, and the results are correct and understandable. 

· Different classification methods are used in the experiments, but there is no specific justification for the choice of these methods in the article.

·  The title of the article and the mention of the use of artificial intelligence are misleading. In this study, based on the information provided, AI-based methods are not used. There are also no arguments that would suggest that AI-based methods should be used.

Author Response

Responses to Reviewer 2

Comment 1.  The aim of this study is to compare the effects of shade drying and sun-drying in the open air on the ventral (upper surface) and dorsal (lower surface) sides of mint leaves. The article presents the results of leaf classification using traditional classification methods. The topic of the article is quite interesting, there is a practical problem that needs to be solved using machine learning methods. However, it is not clear what is the novelty of the article, as classical methods are used to solve the specific problem. The current state of the work is unacceptable, and major changes are needed to improve its quality. The novelty of this study is justified by the fact that „The evaluation of the changes in mint leaves caused by two natural drying techniques was performed using models based on selected texture parameters extracted from the digital color images converted to color channels R, G, B, L, a, b, X, Y, and Z built using machine learning algorithms from groups of Lazy, Rules, and Trees.“ However, this statement is not entirely clear, and an insufficient argument for the novelty of this study. More emphasis should be placed on the novelty of the proposed method (research or solution) and its impact on the field (what new advance or contribution the paper represents). There are no new methods or solutions to specific problems in the article, only data analysis and experimental results are provided.

Response 1. Thank you for your valuable guidance. The previous statement has been changed. The novelty of the study has been justified in more detail.

The article presents a new approach to assessing changes in mint leaves caused by drying techniques. The leaf classification performed using models based on selected texture parameters extracted from the digital color images converted to color channels R, G, B, L, a, b, X, Y, and Z built using machine learning algorithms from groups of Lazy, Rules, and Trees is a great novelty of this study. For the first time, the assessment of the influence of natural drying techniques on the structure of mint leaves was carried out using attributes selected from sets of 1629 image textures from different color channels. Furthermore, monitoring the degree of changes in the structure of leaves under the influence of drying without the need to damage or destroy the leaves can be a problem. The proposed approach may be a non-destructive, objective, cost-effective, and fast practical solution to this problem.

The problem addressed in this study is to reveal the textural changes caused by the drying techniques used to preserve the mint leaf for a long time, with technological solutions. The proposed solution for this is to combine the power of computer vision and artificial intelligence, which are used in many different agricultural applications today [16,17]. In this way, discrimination systems with non-destructive, non-biased, high-accuracy and autonomous capabilities can be developed. The proposed solution also does not depend on a single artificial intelligence algorithm or a single image space. The robustness of the proposed approach is proven with different machine learning methods and different image spaces applied experimentally.

Comment 2.  It is not substantiated that the manuscript is scientifically sound, the experimental design to test the hypothesis is chosen correctly, and all experiments performed are done correctly, and the results are correct and understandable.

Response 2. Thanks for your valuable comments. The revision is as follows. Please also see the revised article

This study aimed to analyze the changes caused by two different drying methods on mint leaves. Because these changes cause significant changes in the nutrient content of the leaf. It is important to provide drying with the least loss and to develop additional techniques for this. In order to clearly observe the effect of drying, the detectability of these changes with computer vision and artificial intelligence algorithms is adopted as a hypothesis. The most obvious changes as a result of drying occur in leaf textures and therefore textures analysis is performed for computer vision. In experimental studies, different image spaces and different machine learning techniques are used to prove that the proposed method is scientifically sound.

 The combination of digital color imaging and artificial intelligence proved to be an effective approach to the evaluation of the effect of natural drying techniques on the mint leaf structure. Models built based on selected textures of images in color channels R, G, B, L, a, b, X, Y, and Z were useful for monitoring the changes in the ventral side (upper surface) and dorsal side (lower surface) of mint leaves caused by shade drying and open-air sun drying. Fresh and dried leaves were distinguished with an accuracy of up to 100% using selected machine learning algorithms from groups of Lazy, Rules, and Trees. The obtained results were promising and the developed procedure involving image features and artificial intelligence could be used to evaluate the quality of mint leaves dried by other techniques. When the accuracy rates obtained as a result of experimental studies are examined, it is seen that the discrimination accuracy in all steps is generally above 90%. Compliance and consistency of all results indicate the accuracy and robustness of the results. Besides traditional machine learning, also deep learning could be applied in further studies for the classification of fresh and dried leaves. Other drying techniques may also be used.

Comment 3.  Different classification methods are used in the experiments, but there is no specific justification for the choice of these methods in the article.

Response 3. Thank you for your attention and interest. It has been specified.

Various machine learning algorithms such as IBk, KStar and LWL from the group of Lazy, JRip and PART from Rules, Bayes Net and Naive Bayes from Bayes, Logistic, Multilayer Perceptron and RBF Classifier from Functions, Random Forest, J48 and LMT from Trees, and Filtered Classifier, Multi Class Classifier and Logit Boost from Meta were tested.

It was found that KStar from the group of Lazy, JRip from the group of Rules, and J48 from the group of Trees were the most effective. These algorithms provided the highest average accuracies of the classification of up to 100% for the selected datasets. It meant completely correct distinguishing the samples. Therefore, the results obtained by the KStar, JRip and J48 were chosen to be presented in this paper.

Comment 4.  The title of the article and the mention of the use of artificial intelligence are misleading. In this study, based on the information provided, AI-based methods are not used. There are also no arguments that would suggest that AI-based methods should be used.

Response 4. Many thanks for your attention. In our study various machine learning algorithms were applied. Machine Learning can be considered a subfield of Artificial Intelligence. Among the AI-based methods, ML algorithms are characterized by the robust discrimination ability of agricultural product features. ML occupies a common area with Artificial Intelligence.

Örnek MN, Örnek HK (2021) Developing a deep neural network model for predicting carrots volume. Food Meas 15:3471–3479.

Reviewer 3 Report

The paper is well structured with clear and objective writing.
The literature review is robust and duly updated, namely with reference scientific publications.
The data and technical results of the study are presented in a detailed and complete manner.
However, the authors should further clarify what the practical contribution of the study is in the field. In practical terms, what is the importance of assessing the quality of dried mint leaves? What effects are intended to be achieved by this study? These questions should be answered objectively and in detail, namely in the Abstract and, above all, in the Introduction and Conclusions chapters.
In the Conclusions chapter, the authors should also present recommendations for future research work.

Author Response

Responses to Reviewer 3

Comment 1. In practical terms, what is the importance of assessing the quality of dried mint leaves? What effects are intended to be achieved by this study? These questions should be answered objectively and in detail, namely in the Abstract and, above all, in the Introduction and Conclusions chapters.

Response 1.   Thank you for the comment. The details have been added to the Abstract, Introduction, and Conclusions as follows:

Abstract

“The developed approach may be used in practice to monitor the changes in the structure of mint leaves caused by drying in a non-destructive, objective, cost-effective, and fast manner without the need to damage the leaves.“

  1. Introduction

“The article presents a new approach to assessing changes in mint leaves caused by drying techniques. The leaf classification performed using models based on selected texture parameters extracted from the digital color images converted to color channels R, G, B, L, a, b, X, Y, and Z built using machine learning algorithms from groups of Lazy, Rules, and Trees is a great novelty of this study. For the first time, the assessment of the influence of natural drying techniques on the structure of mint leaves was carried out using attributes selected from sets of 1629 image textures from different color channels. Furthermore, monitoring the degree of changes in the structure of leaves under the influence of drying without the need to damage or destroy the leaves can be a problem. The proposed approach may be a non-destructive, objective, cost-effective, and fast practical solution to this problem.

The problem addressed in this study is to reveal the textural changes caused by the drying techniques used to preserve the mint leaf for a long time, with technological solutions. The proposed solution for this is to combine the power of computer vision and artificial intelligence, which are used in many different agricultural applications today [16,17]. In this way, discrimination systems with non-destructive, non-biased, high-accuracy and autonomous capabilities can be developed. The proposed solution also does not depend on a single artificial intelligence algorithm or a single image space. The robustness of the proposed approach is proven with different machine learning methods and different image spaces applied experimentally. “

  1. Conclusions

“This study aimed to analyze the changes caused by two different drying methods on mint leaves. Because these changes cause significant changes in the nutrient content of the leaf. It is important to provide drying with the least loss and to develop additional techniques for this. In order to clearly observe the effect of drying, the detectability of these changes with computer vision and artificial intelligence algorithms is adopted as a hypothesis. The most obvious changes as a result of drying occur in leaf textures and therefore textures analysis is performed for computer vision. In experimental studies, different image spaces and different machine learning techniques are used to prove that the proposed method is scientifically sound.”

Comment 2. In the Conclusions chapter, the authors should also present recommendations for future research work.

Response 2.   Thank you for the comment. It has been specified in more detail in the Discussion and Conclusions chapters.

“Our research has set new directions for the application of artificial intelligence in the drying and discrimination of mint. The research carried out with the use of traditional machine learning models may be expanded in the future to include deep learning to distinguish fresh and dried mint leaves.”

“The obtained results were promising and the developed procedure involving image features and artificial intelligence could be used to evaluate the quality of mint leaves dried by other techniques. When the accuracy rates obtained as a result of experimental studies are examined, it is seen that the discrimination accuracy in all steps is generally above 90%. Compliance and consistency of all results indicate the accuracy and robustness of the results. Besides traditional machine learning, also deep learning could be applied in further studies for the classification of fresh and dried leaves. Other drying techniques may also be used.”

Round 2

Reviewer 2 Report

Despite the changes made, the article lacks a detailed description (choice of methods, parameters, detailed algorithm, etc.) and logical reasoning for the actions and procedures involved. It is still not properly substantiated that the manuscript is scientifically sound, the experimental design to test the hypothesis is chosen correctly, and all experiments performed are done correctly, and the results are correct and clear.

The article itself is quite interesting and clearly written. However, it is not clear what the novelty is in this article, as the classical methods are used to solve specific problems.

The article states that in addition to traditional machine learning, deep learning can be applied to classify fresh and dried leaves in future studies. However, as the rather good experimental results show, the need to use deep learning-based methods is not justified.

The article claims that KStar, JRip, and J48 from Trees were the most effective. However, this is a declarative statement, which is not confirmed by experimental results.

A more detailed justification that the proposed methodology and approach to solving this problem is unique or completely new is missing.

The arguments provided by the authors are not sufficient and not entirely correct, which would argue that artificial intelligence was used in this study.

Author Response

RESPONSES TO REVIEWER 2

Comment 1.  Despite the changes made, the article lacks a detailed description (choice of methods, parameters, detailed algorithm, etc.) and logical reasoning for the actions and procedures involved. It is still not properly substantiated that the manuscript is scientifically sound, the experimental design to test the hypothesis is chosen correctly, and all experiments performed are done correctly, and the results are correct and clear.

Response 1. Thank you for your valuable guidance. The following statements about the selection of the method, the algorithm information, and the accuracy of the results have been added to the revised article.

The methods chosen for classification are frequently used for machine learning. Any machine learning method used produces successful results if the extracted features strongly represent the target data. The physical changes caused by drying directly affect the pixel distribution in the image. For this, the texture-based extracted features are strong for distinguishing the drying method. In different spaces, the texture changes are also different. Some spaces show change more clearly. For example, the green channel of the image is used in studies involving the segmentation of the retinal blood vessel. Because in this channel, the blood vessel is more vivid [26]. Therefore, examining the changes in the texture on different channels may provide a more accurate distinction. In addition to the powerful features, the chosen classification algorithm also has a significant effect on success. Although strong features generally provide successful results, different learning methods provide different accuracy due to their methodology. In this context, extracting texture features from different channels and obtaining results with different classification algorithms makes the choice of method in this study powerful. The methodological steps of the study are shown in Figure 5. After all these steps, different result metrics and indicators showing the performance of the study are shared in the result section to show that all the experiments were done correctly. As a result, the confusion matrices with accuracies for each class, the average accuracies and values of Precision, Recall, F-Measure, and MCC (Matthews Correlation Coefficient) were determined [27-30].

Figure 5. Algorithm steps applied for experimental study

Comment 2.  The article itself is quite interesting and clearly written. However, it is not clear what the novelty is in this article, as the classical methods are used to solve specific problems.

Response 2. Thanks for your valuable comments. The novelty of the study is not to introduce a new method, but to show that the changes that occur after drying methods can be distinguished with computer vision and machine learning applications. The following statement indicating the novelty of the study has been added to the revised article.

The robustness of the proposed approach is proven with different machine learning methods and different image spaces applied experimentally. The novelty of the study is to show that the changes that occur after drying can be distinguished with computer vision and machine learning applications.

Comment 3.  The article states that in addition to traditional machine learning, deep learning can be applied to classify fresh and dried leaves in future studies. However, as the rather good experimental results show, the need to use deep learning-based methods is not justified.

Response 3. Thanks for your valuable comments. Deep learning can be used in any application that includes machine learning. It is more recommended to use deep learning, especially if the number of data increases. The figure [1] below explains this. Already, the abundance and diversity of data is very important in learning-based applications. In case of small data, classical machine learning approaches should be used to reduce computational complexity. However, deep learning is indispensable for big data with different types and different samples [2]. In future studies, data with more samples will be studied. For this reason, it is planned that the success will be more reliable and accurate with the use of deep learning. For better understanding, the relevant statement has been revised as follows.

Figure. The performance of deep learning with respect to the amount of data [1]

[1] Alom, M.Z.; Taha, T.M.; Yakopcic, C.; Westberg, S.; Sidike, P.; Nasrin, M.S.; Hasan, M.; Van Essen, B.C.; Awwal, A.A.S.; Asari, V.K. A State-of-the-Art Survey on Deep Learning Theory and Architectures. Electronics 2019, 8, 292. https://doi.org/10.3390/electronics8030292

[2] Guo, Q., Jin, S., Li, M. et al. Application of deep learning in ecological resource research: Theories,     methods, and challenges. Sci. China Earth Sci. 63, 1457–1474 (2020). https://doi.org/10.1007/s11430-019-9584-9

For the deep learning application, data with more samples will be created in the future and experiments will be carried out with different deep learning models on this data. Also, different drying techniques will be included in the dataset.

Comment 4.  The article claims that KStar, JRip, and J48 from Trees were the most effective. However, this is a declarative statement, which is not confirmed by experimental results.

Response 4.  Thanks for your valuable comments. The most effective algorithms in classification success were verified with tables containing complexity matrices and classification performance metrics. The classification accuracies of the machine learning algorithms used in the study are compared in these tables.

Comment 5.  A more detailed justification that the proposed methodology and approach to solving this problem is unique or completely new is missing.

Response 5. Thank you for the comment. The methodology used in this study is not unique, it is not claimed. The novelty of this study is to detect the changes caused by drying techniques on the leaf with machine learning methods. Regarding this, the following information has been added to the revised article.

The robustness of the proposed approach is proven with different machine learning methods and different image spaces applied experimentally. The novelty of the study is not to introduce a new method, but to show that the changes that occur after drying methods can be distinguished with computer vision and machine learning applications.

Comment 6.  The arguments provided by the authors are not sufficient and not entirely correct, which would argue that artificial intelligence was used in this study.

Response 6.  Thank you for the comment. In our article titled “The use of digital color imaging and artificial intelligence for the evaluation of the effects of shade drying and open-air sun drying on the mint leaf quality”, machine learning algorithms are used for classification. Artificial intelligence is a field of science that encompasses machine learning (Figure). However, taking into account your suggestion, we have changed our article title as follows.

“The use of digital color imaging and machine learning for the evaluation of the effects of shade drying and open-air sun drying on the mint leaf quality”

Figure.  The Relationship of AI, ML and DL [1-10]

[1]Amaro Junior, E. (2022). Artificial intelligence and Big Data in neurology. Arquivos de Neuro-Psiquiatria, 80, 342-347.

[2]Li, S., Deng, Y. Q., Zhu, Z. L., Hua, H. L., & Tao, Z. Z. (2021). A Comprehensive Review on Radiomics and Deep Learning for Nasopharyngeal Carcinoma Imaging. Diagnostics, 11(9), 1523.

[3]Ryu, J. Y., Chung, H. Y., & Choi, K. Y. (2021). Potential role of artificial intelligence in craniofacial surgery. Archives of Craniofacial Surgery, 22(5), 223.

[4]Abdellah, A., & Koucheryavy, A. (2020). Survey on artificial intelligence techniques in 5G networks. J. Inf. Technol. Telecommun. SPbSUT Russ, 8, 1-10.

[5]Sindhu, V., Nivedha, S., & Prakash, M. (2020). An empirical science research on bioinformatics in machine learning. Journal of Mechanics Of Continua And Mathematical Sciences.

[6]Rozlosnik, A. E. (2020, June). Reimagining infrared industry with artificial intelligence and IoT/IIoT. In Thermosense: Thermal Infrared Applications XLII (Vol. 11409, pp. 233-257). SPIE.

[7]Adamu, J. A. (2019). Advanced Stochastic Optimization Algorithm for Deep Learning Artificial Neural Networks in Banking and Finance Industries. Risk and Financial Management, 1(1), p8-p8.

[8]Kusunose, K., Haga, A., Abe, T., & Sata, M. (2019). Utilization of artificial intelligence in echocardiography. Circulation Journal, CJ-19.

[9]Latif, J., Xiao, C., Imran, A., & Tu, S. (2019, January). Medical imaging using machine learning and deep learning algorithms: a review. In 2019 2nd International conference on computing, mathematics and engineering technologies (iCoMET) (pp. 1-5). IEEE.

[10]Campesato, O. (2020). Artificial intelligence, machine learning, and deep learning. Mercury Learning and Information.

Reviewer 3 Report

The authors responded well to suggestions and questions, bridging the identified gaps.

Author Response

Thank you very much for this comment